# An Innovative Compact Split-Ring-Resonator-Based Power Tiller Wheel-Shaped Metamaterial for Quad-Band Wireless Communication

**DOI:** 10.3390/ma16031137

**Published:** 2023-01-28

**Authors:** Md. Salah Uddin Afsar, Mohammad Rashed Iqbal Faruque, Sabirin Abdullah, Mohammad Tariqul Islam, Mayeen Uddin Khandaker, K. S. Al-Mugren

**Affiliations:** 1Space Science Centre (ANGKASA), Institute of Climate Change (IPI), Universiti Kebangsaan Malaysia, Bangi 43600, Malaysia; 2Department of Electrical, Electronic & Systems Engineering, Faculty of Engineering & Built Environment, Universiti Kebangsaan Malaysia, Bangi 43600, Malaysia; 3Centre for Applied Physics and Radiation Technologies, School of Engineering and Technology, Sunway University, Bandar Sunway 47500, Malaysia; 4Department of Physics, College of Science, Princess Nourah bint Abdulrahman University, Riyadh 11671, Saudi Arabia

**Keywords:** ENG metamaterial, TE mode, polarization-insensitive, wireless communication

## Abstract

A split-ring resonator (SRR)-based power tiller wheel-shaped quad-band ℇ-negative metamaterial is presented in this research article. This is a new compact metamaterial with a high effective medium ratio (EMR) designed with three modified octagonal split-ring resonators (OSRRs). The electrical dimension of the proposed metamaterial (MM) unit cell is 0.086λ × 0.086λ, where λ is the wavelength calculated at the lowest resonance frequency of 2.35 GHz. Dielectric RT6002 materials of standard thickness (1.524 mm) were used as a substrate. Computer simulation technology (CST) Microwave Studio simulator shows four resonance peaks at 2.35, 7.72, 9.23 and 10.68 GHz with magnitudes of −43.23 dB −31.05 dB, −44.58 dB and −31.71 dB, respectively. Moreover, negative permittivity (ℇ) is observed in the frequency ranges of 2.35–3.01 GHz, 7.72–8.03 GHz, 9.23–10.02 GHz and 10.69–11.81 GHz. Additionally, a negative refractive index is observed in the frequency ranges of 2.36–3.19 GHz, 7.74–7.87 GHz, 9.26–10.33 GHz and 10.70–11.81 GHz, with near-zero permeability noted in the environments of these frequency ranges. The medium effectiveness indicator effective medium ratio (EMR) of the proposed MM is an estimated 11.61 at the lowest frequency of 2.35 GHz. The simulated results of the anticipated structure are validated by authentication processes such as array orientation, HFSS and ADS for an equivalent electrical circuit model. Given its high EMR and compactness in dimensions, the presented metamaterial can be used in S-, C- and X-band wireless communication applications.

## 1. Introduction

Metamaterial is a congress of non-natural physical structures designed to achieve advantageous and uncommon electromagnetic properties. The effective properties of metamaterials are defined and measured in terms of permittivity (ε) and permeability (μ) [1,2]. A hypothetical ℇ-negative and µ-negative metamaterial termed DNG or LHM metamaterial was introduced in 1968 by Russian physicist Victor Veselago [3]. The unique properties of this metamaterial have drawn the attention of scientists all over the world for various applications in the microwave frequency range [4,5,6]. Nowadays, microwave-based applications are used in filtering [7], hidden cloaking [8], SAR reduction [9], absorber design [10], bandwidth enhancement [11], etc. A unit cell itself cannot acts as a complete metamaterial but is a systematic periodic array of metal–dielectric–metal or dielectric–metal structure upon a host substrate [12]. An S-shaped metamaterial with an EMR of 4.8 was designed for sensing applications in the microwave range [13]. A dual-band flexible metamaterial was designed on a nickel aluminate (NiAl2O4) substrate with a 42% aluminum concentration and dimensions of 12.5 × 10 mm^2^, covering the X and Ku bands [14]. Recently, a metamaterial was reported that a contained rectangular-shaped SRR. This metamaterial was utilized to sense concrete, temperature and humidity [15]. Islam et al. in [16] introduced a SNG metamaterial that shows triple band resonance for microwave application. Moreover, Smith et al. proposed a three-dimensional metamaterial built on reedy wire, along with a split-ring resonator [17]. In numerical simulation, the MM exhibited a double-negative characteristic with a wideband spectrum. A triple-band polarization-dependent MM with dimensions of 8 × 8 mm^2^ was designed on an RT6002 substrate and yielded at 0.92 GHz, 7.25 GHz and 14.83 GHz, covering the S, C and Ku bands [18]. A tri-band MM with dimensions of 10 × 10 mm^2^ and a Greek key shape was designed on an RT 5880 dielectric. In numerical simulation, it showed triple resonance peaks at 2.40, 3.50 and 4.0 GHz [19]. An epsilon-negative, delta-shaped metamaterial comprising an SSR (square ring resonator) exhibited tri-resonance crests that covered the C and X bands [20]. Another triple-band metamaterial with dimensions of 5 × 5 mm^2^ was presented by Liu et al. in 2016 [21] with an RCER (reformed circular electric resonator). This MM with a low (5.45) effective medium ratio (EMR) was resonant in the frequency ranges of 9.70 GHz to 10.50 GHz and 15 GHz to 15.70 GHz.

A different metamaterial with a pie-shaped metallic resonator surrounded by an SRR was presented in [22]. This tri-band MM was designed on a substrate with dimensions of 8 × 8 mm^2^ and covered the microwave S, C and X bands. An SRR-based triple-band metamaterial was designed with a double circular ring [23]. This multiunit cell-based MM was resonant at 5.6 GHz of Wi-MAX and 2.45 of WLAN. In 2019, Almutairi et al. [24] designed a metamaterial based on a CSRR (complementary split-ring resonator) with dimensions of 5 × 5 mm^2^. It showed resonance at 7.5 GHz with an EMR of 8. Moreover, an SNG metamaterial with dimensions of 5 × 5 × 1 mm^3^ which comprising a concentric ring, along with a cross line, was designed on an FR-4 substrate [25]. It exhibited dual resonance peaks at 13.9 GHz and 27.5 GHz and was used to enhance the performance of a microstrip transmission line. A metamaterial was designed on an elliptical graphene nanodisk with a periodic pattern on a thin SiO_2_ dielectric layer, as reported in [26]. Recently, two ceramic dielectrics were synthesized using MGa_2_O_4_ (M = Ca, Sr) and LiF for to enhance the gain and performance of antennae [27,28]. An MM was designed using critical coupling at the gaps of two SRRs for total broadband transmission electromagnetic (EM) waves in a C-band application [29]. A cadmium sulfide (CdS) nanocrystalline coating with conducting polyaniline was designed to synthesized polyaniline-coated CdS nanocomposites characterized by UV–vis absorption [30]. In 2022, Amali et al. designed a nanocomposite using a potentiostatic method, which offered excellent electrocatalytic activity for nitrite oxidation [31].

In his research article, we present a new metamaterial that is an aggregation of three modified octagonal rings, along with a split-ring resonator. This power tiller wheel-shaped MM is compact in size, with an EMR of 11.61. In numerical simulation, it exhibits quad-band resonance peaks at 2.35, 7.72, 9.23 and 10.68 GHz, covering the S, C and X bands. Moreover, it also exhibits negative permittivity (ε) and a negative refractive index (n), with simultaneous near-zero permeability (µ). Such characteristics can be applied to comprehend various electronic components with different features and utilities. The main aim of this simple but first-hand design is to target versatile uses in wireless communication. The simulated results are verified by validation processes, confirming the reliability, consistency and efficiency of the proposed metamaterial. The ADS simulated result using a circuit model and Ansys HFSS 3D high-frequency software (high-frequency structure simulator) results show excellence harmony with the CST results.

## 2. Design Parameters of the Metamaterial and Simulation Setup

Figure 1a shows the front view of the unit cell, which is labeled with symbols. It is a new combination of three different octagonal rings surrounded by a split-ring resonator (SRR). A popularly used dielectric Rogers RT 6002 with dimensions 11 × 11 mm^2^ and a thickness of 1.524 mm is used as a substrate. The dielectric constant, thermal conductivity and tangent loss of RT6002 are 2.94, 0.6 W/m/K and 0.0012, respectively. Copper (annealed) with an electrical conductivity of 5.96 × 10^7^ Sm**^−^**^1^ is used for all resonators of the upper layer. The outer and inner radii of the first octagon are R_1_ = 4.3 mm and R_2_ = 3.8 mm, respectively, whereas the radii of the second octagon are r_1_ = 3.3 mm and r_2_ = 2.8 mm, respectively. Each split gap (g) of the octagon is 0.40 mm. The outer and inner radii of the smallest octagon are r_3_ = 1.5 mm and r_4_ = 0.75 mm, respectively. These three octagons (OSRRs) are placed at the center an SRR with dimensions of 10.40 × 10.40 mm^2^ and a split gap (G) of 0.50 mm. The three octagons are attached to each other by four metal strips with a length of 3 mm and a width of 0.40 mm.

It is noteworthy that the width of the SRR (t), as well as that of the first two octagons, is 0.50 mm, whereas the width of the smallest octagon (e) is 0.75 mm. The perspective view and the simulation setup of the proposed MM are depicted at Figure 1b,c respectively. The symbolic presentations of the design parametric values of the projected unit cell are given in the Table 1.

Proper boundary conditions are applied to attain the expected results from the proposed metamaterial design. The electromagnetic radiation propagates along the z coordinate, whereas the perfect electric conductor (PEC) and the perfect magnetic conductor (PMC) propagate along the x coordinate and y coordinate, respectively.

### 2.1. Extraction Process of Medium Parameters

To extract the various properties of the material, the S-parameters model of the post-processing module of CST can be deployed [32]. This software is applied to obtain information associated with the three important characteristics of permittivity ($\varepsilon_{r}$), permeability (${µ}_{r}$) and refractive index ($n_{r}$) of the proposed unit cell of metamaterial to realize EM properties [33]. Moreover, the refractive index, S parameters (reflection and transmission coefficient) and impedance can be correlated with the help of Equations (1)–(5) of the robust retrieval method described in [34].
(1)$$\left|S_{11}\right|=\frac{R_{01}-R_{01}e^{2nkd}}{1-R_{01}^{2}e^{i2nkd}}$$
(2)$$\left|S_{21}\right|=\frac{(1-R_{01}^{2}{)}_{\;}e^{i2nkd}}{1-R_{01}^{2}e^{i2nkd}}$$
(3)$$R_{01}=\frac{Z^{2}-1}{{\left(Z+1\right)}^{2}}$$

Here, impedance is expressed as:(4)$$Z=\pm \surd \left\{\frac{{\left(1+S_{11}\right)}^{2}-S_{12}^{2}}{{\left(1-S_{11}\right)}^{2}-S_{12}^{2}}\right\}$$
where $S_{11}$ = reflection coefficient, and $S_{21}$ = transmission coefficient. Then,
(5)$$\mathrm{Refractive}\;\mathrm{Index},\;n_{r}=\frac{1}{kd}\left[{\left\{\ln\left(e^{inkd}\right)\right\}}^{\prime \prime }+2m\pi -i{\left\{ln\left(e^{inkd}\right)\right\}}^{\prime }\right]$$

The electromagnetic wave is fixed to propagate along the z direction, together with the perfect electric and the magnetic fields, which are applied as boundary conditions along the x and y directions, respectively. Additionally, the relative permittivity $\left(\varepsilon_{r}\right)$ and relative permeability (${µ}_{r}$) can be derived from Equations (6) and (7), respectively, using the Nicolson–Ross–Weir (NRW) technique [35].
(6)$$\mathrm{Permittivity},\;\varepsilon_{r}=\frac{c}{j\pi fd}\;.\left[\frac{1}{1+\left(S_{21}+S_{11}\right)}-\frac{\left(S_{21}+S_{11}\right)}{1+\left(S_{21}+S_{11}\right)\;}\right]$$
(7)$$\mathrm{Permeability},\;{µ}_{r}=\frac{c}{j\pi fd}\;.\left[\frac{1}{1+\left(S_{21}+S_{11}\right)}-\frac{\left(S_{21}-S_{11}\right)}{1+\left(S_{21}+S_{11}\right)\;}\right]$$
where c = speed of light, f = frequency and d = the thickness of the substrate. MATLAB codes are written based on Equations (6) and (7). The values of the material parameters extracted through the NRW technique are verified and compared with the results of numerical simulation.

### 2.2. Parametric Studies

#### 2.2.1. Design Hierarchy

The chronological development of the proposed metamaterial unit cell is shown in the Figure 2. The design architecture and its morphology are set up to achieve the highest performance possible. An iterative method is applied to record feedback of the unit cell in order to determine the transmission coefficient (S_21_). Design 2(a) comprises a split-ring resonator (SRR) along with an octagonal ring on the substrate layer. It yields resonance at 2.44 GHz, 8.67 GHz and 10.84 GHz. Another comparatively smaller octagonal ring of the same width is added to the first design, which is shown in the Figure 2b. In CST simulation, it exhibits quad-band resonance peaks at 2.51 GHz, 8.55 GHz, 9.50 GHz and 11.02 GHz. Again, to test the enhancement of the bandwidths, a small octagon with a width of 0.75 mm is placed at the center of the previous structure, which is shown in Figure 2c, with resonance at 2.50 GHz, 8.57 GHz, 9.49 GHz and 11.03 GHz. Moreover, as shown in Figure 2d, all the octagons are attached to one another with four metal bars with a length of 3 mm and a width 0.40 mm, exhibiting resonance at 2.51 GHz, 8.58 GHz and 11.05 GHz. Finally, four split gaps with a width of 0.40 mm are made in the three octagons, as shown in the Figure 2e. This reformed design yields quad-band resonance at 2.35 GHz, 7.72 GHz, 9.23 GHz and 10.68 GHz with magnitudes of −43.23, −31.05, −44.58 and −31.71 dB, covering the S, C and X bands with excellent bandwidths. This quad-band metamaterial also exhibits negative permittivity (ℇ), a negative refractive index and near-zero permeability (µ). It is noteworthy that RT 6002 dielectric is used as a substrate for all the designs shown in Figure 2a–e. Therefore, the structure shown in Figure 2e is selected as a final unit cell for the proposed MM. The simulated results of the S_21_ from Figure 2a–e are shown in the Table 2. Figure 3 shows the numerical results of S_21_ for all design steps.

#### 2.2.2. Effect of Substrate Materials

Proper dielectric selection is an important task for any metamaterial design. An investigation is conducted to observe the response of different substrate materials. Commercially available flame-retardant FR-4 material, along with two Rogers dielectrics, RT 5880 and RT 6002, are taken into consideration. Three individual substrates are simulated by keeping the resonator structure unchanged. First, dielectric FR-4 shows resonance at 2.04 GHz, 6.66 GHz, 7.97 GHz and 9.42 GHz, with very low magnitudes. Secondly, Rogers RT 5880 yields triple-band resonance peaks at 3.7 GHz, 8.67 GHz and 11.33 GHz, whereas RT6002 shows quad-band resonance peaks at 2.35 GHz, 7.72 GHz, 9.23 GHz and 10.68 GHz, with satisfactory magnitudes and moderate bandwidths. The simulated results cover the S, C and X bands. The transmission coefficients ($S_{21}$) for the three substrate materials are shown in Figure 4.

#### 2.2.3. Unit Cell Dimension Optimization

Various sizes of unit cell for the same dielectric (RT6002) and for the same metal (copper annealed) are inspected to select the appropriate size of the proposed metamaterial. First, the unit cell is simulated with substrate dimensions of 13 × 13 × 1.524 mm^3^, exhibiting quad-band resonance at 2.58 GHz, 7.60 GHz, 9.47 GHz and 10.36 GHz. Secondly, it is simulated with a unit cell with dimensions of 12 × 12 × 1.524 mm^3^, showing quad-band resonance peaks with a small decrement of resonance frequencies. Lastly, it is simulated for dimensions of 11 × 11 × 1.524 mm^3^, showing quad-band resonance at 2.35 GHz, 7.72 GHz, 9.23 GHz and 10.68 GHz, with a better progression of bandwidths. Figure 5 demonstrates the simulated results for the selected sizes of the unit cell.

#### 2.2.4. The Effect of Field Propagation Direction

A change in transmission coefficient (S_21_) is observed with varying electric field and magnetic field direction. Figure 6 demonstrates the simulation setup for changing the field propagation. Initially, electric field (Ex) propagates along the X direction, and the magnetic field (Hy) is applied to the Y direction. The simulation result shows quad-band resonance peaks at 2.35 GHz, 7.72 GHz, 9.23 GHz and 10.68 GHz. If the fields are interchanged with each other, the simulated results show two resonance peaks at 4.58 GHz and 8.38 GHz. Figure 7 illustrates the simulated results for propagation in the ExHy and HxEy directions.

### 2.3. Analysis of Electromagnetic Field and Surface Current

The upper layer of the proposed metamaterial unit cell contains resonant assemblies composed of split gaps and metallic conductors. The split gaps and conductors play the roles of capacitors and inductors, respectively. Electromagnetic force is exerted on the resonators due to the interaction between time-varying EM fields and the unit cell. The induction current flows from one resonator to another through the capacitive split gaps, which are smaller than the wavelength of the incident EM wave. Produced electric and magnetic moments influence the transmission ability and change the material characteristics such as permittivity and permeability. The surface current distribution of the presented MM is illustrated in Figure 8, predicting that at a low-resonance frequency of 2.35 GHz, the outer ring subsidizes a higher amount of current. At the lower frequency, the inductive reactance is also low because the outer ring contributes a low impedance route. A significant amount of current flow decreases in the first outer ring at the second resonance frequency of 7.72 GHz because an increase in impedance occurs with the increase in frequency. At this frequency, non-uniform and random movement of current is detected in bars connecting the octagons, which eventually reduces the overall current flow. In the two inner octagons, current flow is reduced because of the neutralization of two opposite flows. For the same reasons, current flow becomes insufficient at a resonance frequency of 9.23 GHz, and high current flow is observed through the edges of all horizontal sides of all rings compared to the previous two positions. It is also noteworthy that a substantial amount of current is contributed by the two horizontal sides of the outer ring, owing to lower impedance applied by the split gap at a resonance frequency of 10.68 GHz.

Time-varying charge flow is mainly responsible for generating magnetic field according to the Amperes law in association with Faraday’s law of induction, which, in turn, produces an electric field due to electromagnetic interaction [36]. The induced E field and H field can be inspected using the Maxwell’s curl Equations (8)–(12), as presented in [37].

Horizontal magnetic field component:(8)$$\nabla \times H=J+\frac{\partial D}{\partial t}$$

Produced electric field related to the magnetic field:(9)$$\nabla \times E=\frac{\partial H}{\partial t}\;$$
where the vector operator is expressed as:(10)$$\nabla =\hat{i}\frac{\partial x}{\partial t}+\hat{j}\frac{\partial y}{\partial t}+\hat{k}\frac{\partial z}{\partial t}$$

Equations (8) and (9) are not sufficient to explain why the two fields interact with materials. Two more equations are required to overcome these limitations [38].
(11)$$\varepsilon \;\left(t\right)=\frac{D\left(t\right)}{E\;\left(t\right)}$$
(12)$$\mu \;\left(t\right)=\frac{B\left(t\right)}{H\left(t\right)}$$

The material properties of permittivity (ε) and permeability (µ) in Equations (11) and (12) are complex parameters and real in the case of isotropic lossless material. A vivid observation magnetic field (H) and electric field (E) for the four resonance frequencies (2.35, 7.72, 9.23 and 10.68 GHz) are illustrated in Figure 9 and Figure 10, respectively. The intensity of the magnetic field and polarity depend on the amount of current and its flow direction. The H-field distribution in Figure 9 shows that at locations in the unit cell where the current density is high, the magnetic field is also high. As shown by the patterns of H-field and E-field distribution in Figure 9 and Figure 10, if the magnetic field changes towards an increment, then the electric field changes inversely. The changing tendency of magnetic and electric fields is determined according to Equations (8) and (9). Furthermore, as every split gap of the unit cell of the proposed MM acts as a capacitor, the electric field intensity in the split gap is increased.

### 2.4. Equivalent LC Circuit of the Unit Cell

An estimated electrical equivalent circuit is drawn and executed by ADS to validate the CST results of the proposed metamaterial. The unit cell is designed with a combination of both metal strips and some split gaps. Every metal strip represents a conductor, whereas every split gap represents a capacitor [39]. In the microwave band, metallic conductor copper can be treated as a perfect conductor that can ignore the ohmic losses [40]. Therefore, the whole unit cell is represented by an LC resonance circuit. The inductance and the capacitance are the main parameters of an LC circuit, denoted by (L) and (C), respectively. Using these two parameters, resonance frequency (f) can be calculated by applying Equation (13).
(13)$$f=\frac{1}{2\pi }\frac{1}{\surd \left(LC\right)}$$

The quasi-state theory can be applied to measure the capacitance within a distance or in a split gap in a circuit [41].
(14)$$C=\varepsilon_{0}\varepsilon_{r}\frac{A\;\left(F\right)}{d}$$
where $\varepsilon_{o}$ is the permittivity in free space, $\varepsilon_{r}$ is the relative permittivity, A is the cross-sectional area of the conducting strip and d is the split gap.

The inductance of a rectangular metal bar can be calculated according to Equation (15) [42].
(15)$$L\left(nH\right)\;\frac{1}{l}=2\times 10^{-4}\left[\ln\left(\frac{l}{w+t}\right)+1.193+0.02235\left(\frac{w+t}{l}\right)\right]K_{G}$$
where $K_{G}$ is the correction factor, *w* is the width, l is the length and t is the thickness of the strip.

An equivalent LC circuit of the proposed MM is illustrated in Figure 11. The whole equivalent circuit comprising eleven (L1 to L11) inductors and twelve (C1 to C12) capacitors is simulated by ADS software. The first resonator on the upper layer is a split-ring resonator (SRR), as indicated by (L1, C1) and (L2, C2), contributing the first resonance frequency of 2.39 GHz, whereas (L3, C3) and (L4, C4) are used for the first octagon that belongs to 7.23 GHz. On the contrary, the second octagon is replaced by (L5, C5) and (L6, C6), which partially contribute to the frequency of 9.21 GHz. C7 and C8 are the coupling capacitors. The joining metal bars and associate gaps are represented by (L9, C10) and (L10, C11), whereas L11 is used for the small central octagon. These components are jointly responsible for the resonance frequency of 10.72 GHz. A comparison between two transmission coefficients that are determined by CST and ADS is shown in Figure 12.

## 3. Results and Discussion

CST microwave studio is deployed to simulate the proposed metamaterial in the frequency range 1–14 GHz. Figure 13 demonstrates the scattering parameters (reflection and transmission coefficients). Numerical simulation yields four resonance frequencies at 2.35, 7.72, 9.23 and 10.68 GHz with magnitudes of −43.23 dB, −31.05 dB, −44.58 dB and −31.71 dB, respectively. These frequency bands cover the S, C, and X bands. Moreover, the response for the reflection coefficient $\left(S_{11}\right)$ shows at 3.33 GHz, 7.92 GHz and 10.39 GHz with magnitudes of −36.30 dB, −13.84 dB and −15.23dB, respectively. It is evident that every resonance of the transmission coefficient $\left(S_{21}\right)$ is followed by bandwidths of 0.36 GHz, 0.46 GHz, 1.42 GHz and 0.30 GHz at the concerned S, C, and X bands, respectively. It is also evident that each resonance of the transmission coefficient $\left(S_{21}\right)$ is tracked by a reflection coefficient $\left(S_{21}\right)$ minimum. Subsequently, the frequency of each value of $S_{21}$ minimum is always lower than the concerned value of the $S_{11}$ minimum frequency. In this regard, it can be concluded that every resonance can be treated as electrical resonance in the proposed metamaterial unit cell [43]. The permittivity (ε), permeability (µ) and refractive index (n) extracted by applying the RRM (robust retrieval method) in CST and NRW in MATLAB are shown in Figure 14, Figure 15 and Figure 16, respectively. Figure 14 shows that the permittivity of the designed MM varies from the positive value to the negative. Again, the values of $S_{21}$ begin when the magnitudes of permittivity fluctuate from maximum to minimum values. Moreover, the starting positive minimum value of µ occurs at the minimum resonance frequency shown in Figure 15. This tendency continues over the whole resonance frequency range. A graph of the refractive index is presented in Figure 16, which reveals negative refractive indices in the frequency ranges of 2.36–3.19 GHz, 7.74–7.87 GHz, 9.26–10.33 and 10.70–11.81 GHz. The negativity of refractive index is a function of frequency that can be utilized to increase the gain and directivity of antennae, whereas the ℇ-negative property is deployed to enhance the bandwidth [44,45]. Finally, a brief comparison is shown on the basis of some important parameters in Table 3.

### 3.1. Array Metamaterial Results

Different types of array combinations are also simulated to test the coupling effect and to verify the consistency of the results, which is the best way to achieve the expected electromagnetic features. Arrays of the proposed MM with dimensions of 1 × 2 and 2 × 2 are shown in Figure 17. These two designs are simulated by the CST and the reflection coefficient (S_11_), and transmission coefficient (S_21_) results are presented in Figure 18. The variations of resonance frequencies among the unit cell and the 1 × 2 and 2 × 2 arrays are given in Table 4, which confirms the consistency of the results.

### 3.2. Validation Using HFSS

In order to verify the reliability and consistency of the performance of the proposed MM, the CST result for the transmission coefficient (S_21_) is authenticated by Ansys HFSS. The simulated result obtained with this software also shows quad-band resonance peaks, with amplitudes remaining nearly unchanged and yielding four resonance peaks at 2.37 GHz, 7.66 GHz, 9.20 GHz and 11.0 GHz with amplitudes of −40.71 dB, −33.08 dB, −44.70 dB and −30.36 dB, respectively. For comparison, the results are shown in Figure 19, in agreement with the CST result.

## 4. Conclusions

In this research article, a quad-band power tiller wheel-shaped ENG metamaterial for S-, C- and X-band applications is presented. The size of the proposed MM unit cell has dimensions of 10 × 10 × 1.524 mm^3^ and is based on an RT6002 dielectric substrate. CST microwave studio is used to simulate the unit cell, showing quad-band resonance peaks at 2.35 GHz, 7.72 GHz, 9.23 GHz and 10.68 GHz with amplitudes of −43.23 dB −31.05 dB, −44.58 dB and −31.71 dB, respectively. The simulated results are also validated by validation processes such as an equivalent electrical circuit model, high-frequency simulation software (HFSS) and various array orientations. The response and contribution of various resonators of the unit cell are inspected by analyzing the E-field, H-field and surface current distribution for propagated electromagnetic radiation. The important features of permittivity, permeability and the refractive index of the metamaterial are extracted using MATLAB. The EMR of the proposed MM is 11.61, which indicates its reliability. The calculated value of λ4 is less than the length (L) of the unit cell, highlighting the compactness of the size of the unit cell. This innovative MM can be deployed to enhance the efficiency of different microwave devices, owing to its NRI and epsilon-negative characteristics. Moreover, the S, C- and X bands are recurrently used for satellite and radar applications.

## Figures and Tables

**Figure 1 materials-16-01137-f001:**
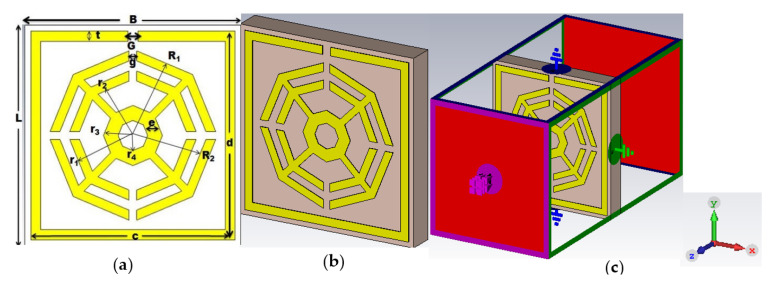
(**a**) Unit cell (**b**) perspective view of the unit cell (**c**) simulation setup.

**Figure 2 materials-16-01137-f002:**
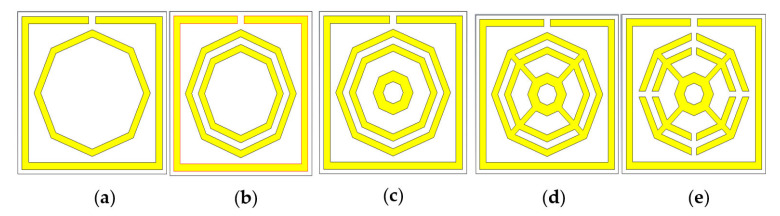
Design optimization views: (**a**) Design-1; (**b**) Design-2; (**c**) Design-3; (**d**) Design-4; (**e**) final design.

**Figure 3 materials-16-01137-f003:**
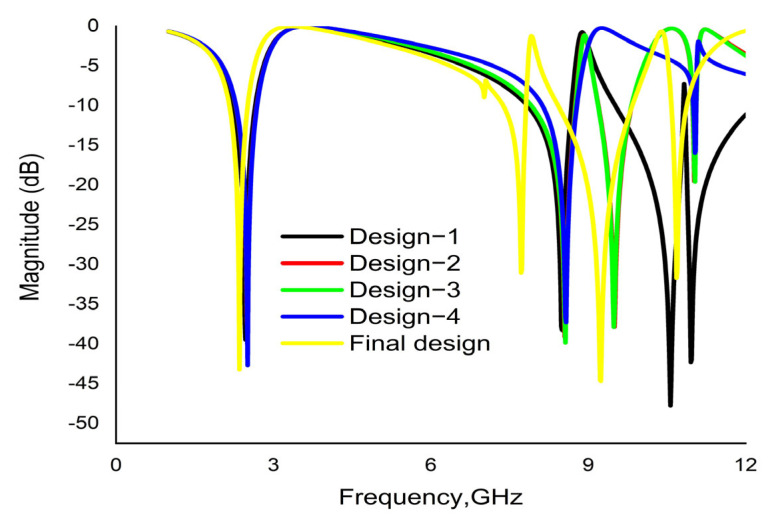
CST result of transmission coefficient (S_21_) for various design steps.

**Figure 4 materials-16-01137-f004:**
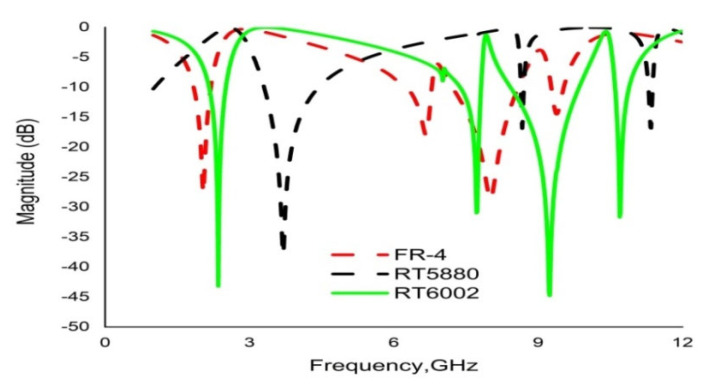
Transmission coefficients ($S_{21}$) for different substrate dielectric materials.

**Figure 5 materials-16-01137-f005:**
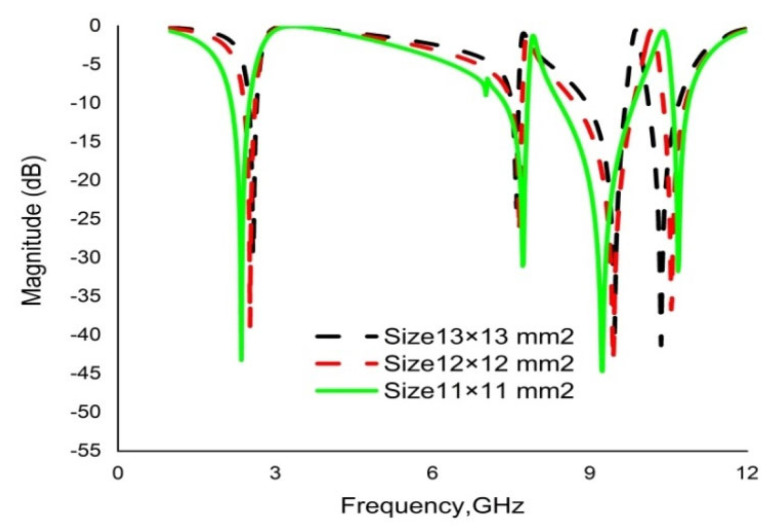
Transmission coefficients ($S_{21}$) for different sizes of unit cell substrate.

**Figure 6 materials-16-01137-f006:**
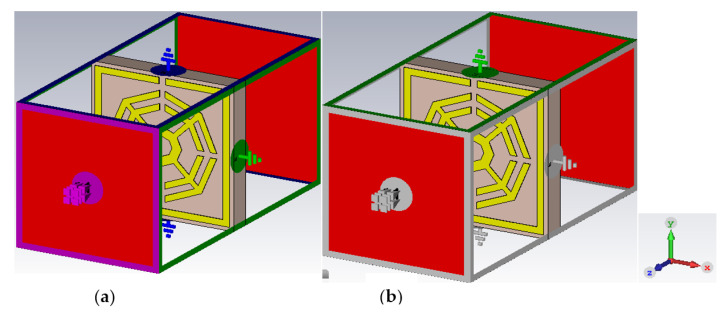
Application of changing electric and magnetic field directions: (**a**) ExHy; (**b**) EyHx.

**Figure 7 materials-16-01137-f007:**
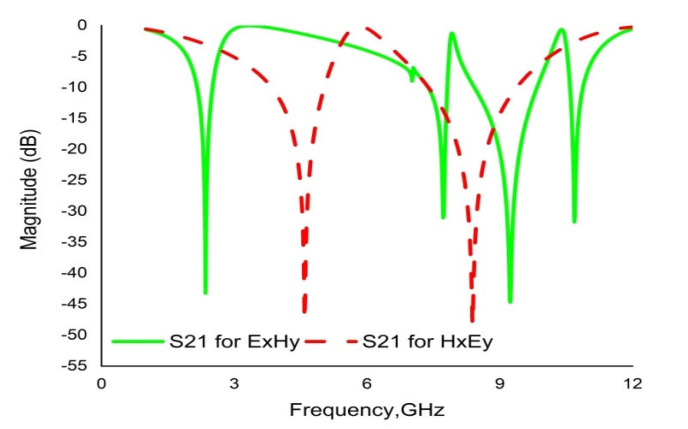
Transmission coefficient ($S_{21}$) with interchanging field propagation.

**Figure 8 materials-16-01137-f008:**
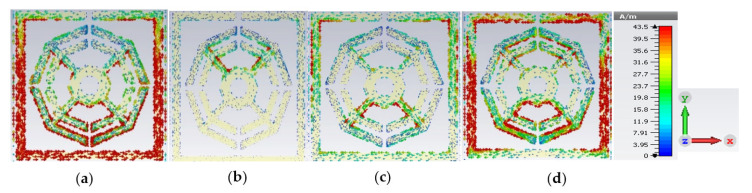
Surface current distribution at (**a**) 2.35 GHz, (**b**) 7.72 GHz, (**c**) 9.23 GHz and (**d**) 10.68.

**Figure 9 materials-16-01137-f009:**
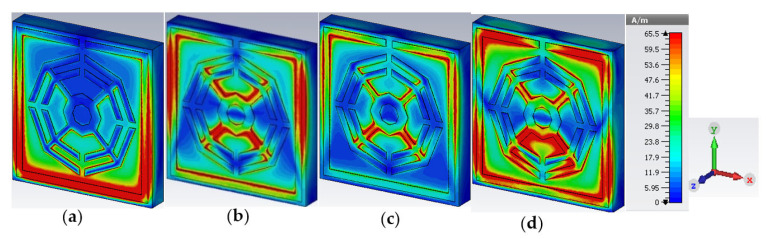
Magnetic field distribution at (**a**) 2.35 GHz (**b**) 7.72 GHz (**c**) 9.23 GHz and (**d**) 10.68 GHz.

**Figure 10 materials-16-01137-f010:**
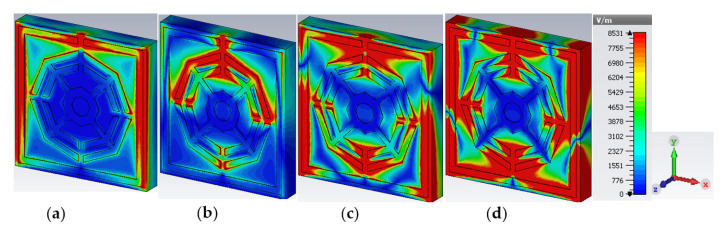
Electric field distribution at (**a**) 2.35 GHz (**b**) 7.72 GHz (**c**) 9.23 GHz and (**d**) 10.68 GHz.

**Figure 11 materials-16-01137-f011:**
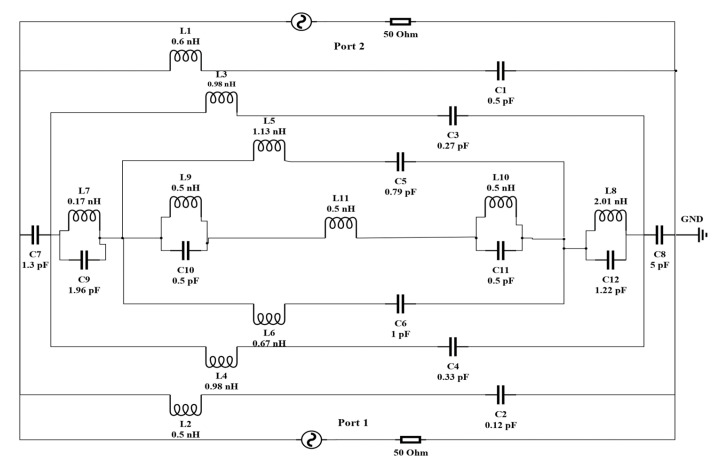
Equivalent electric al circuit model of the proposed metamaterial unit cell.

**Figure 12 materials-16-01137-f012:**
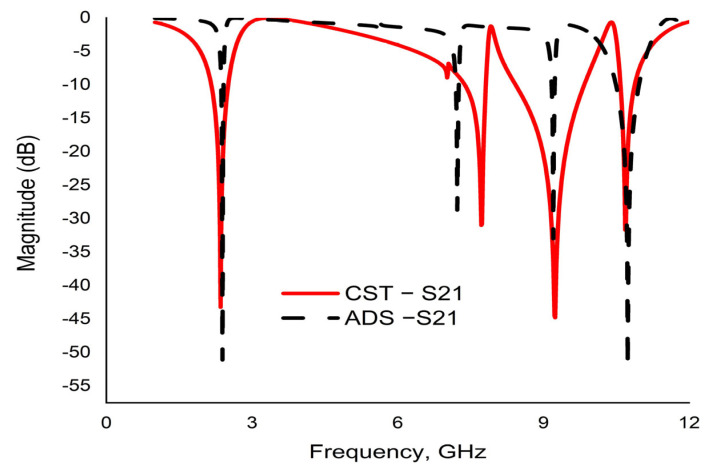
The transmission coefficient ($S_{21}$) of the proposed unit cell simulated by CST and ADS software.

**Figure 13 materials-16-01137-f013:**
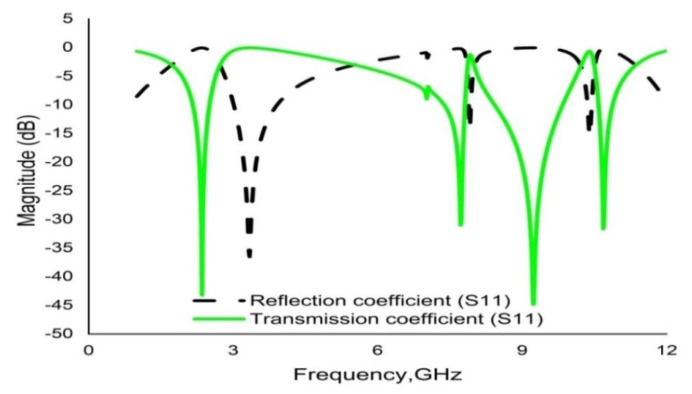
Graph of the S parameters ($S_{11}$ and $S_{21}$) of the unit cell.

**Figure 14 materials-16-01137-f014:**
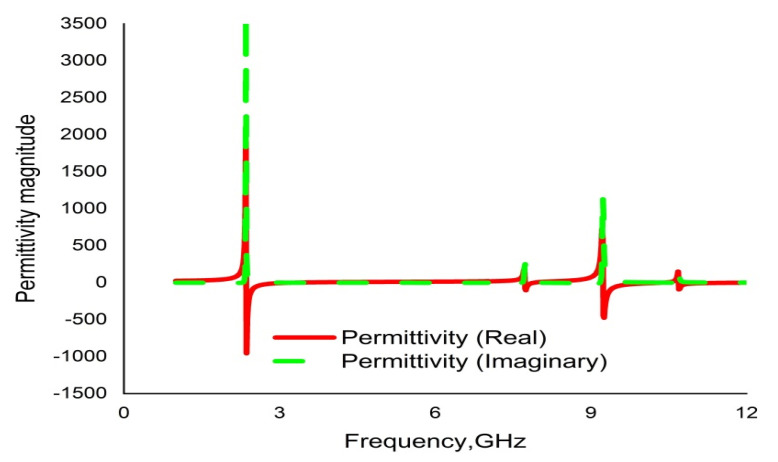
Graph of real and imaginary magnitudes of permittivity (ℇ) for the unit cell.

**Figure 15 materials-16-01137-f015:**
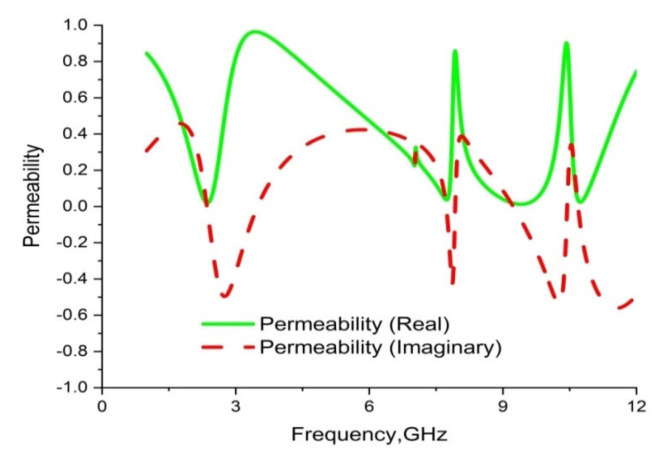
Graph of real and imaginary magnitudes of permeability (µ) for the unit cell.

**Figure 16 materials-16-01137-f016:**
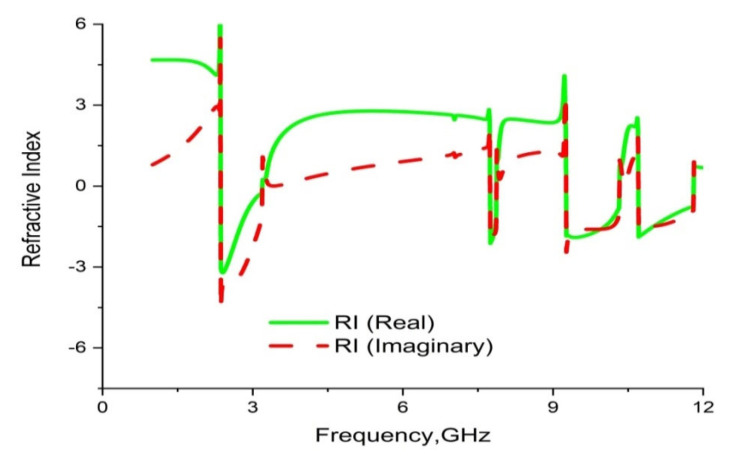
Graph of real and imaginary magnitudes of refractivity for the unit cell.

**Figure 17 materials-16-01137-f017:**
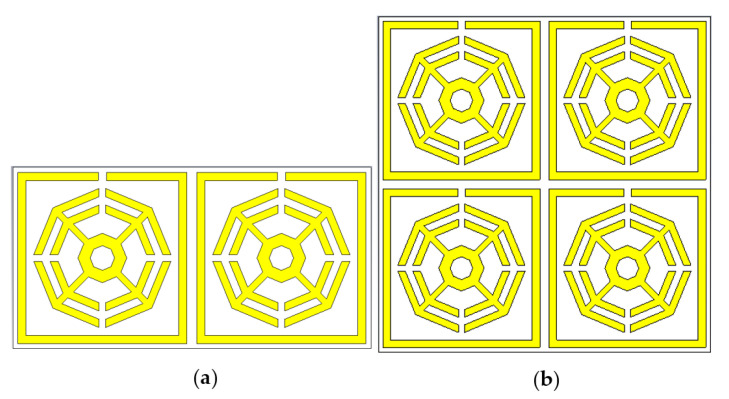
Array combinations: (**a**) 1 × 2 array; (**b**) 2 × 2 array.

**Figure 18 materials-16-01137-f018:**
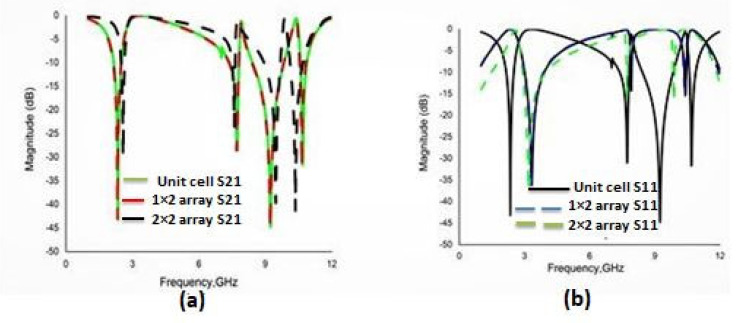
Comparison of S parameters for a array and unit cell (**a**) transmission coefficient $(S_{21}$) and (**b**) reflection coefficient ($S_{11}$).

**Figure 19 materials-16-01137-f019:**
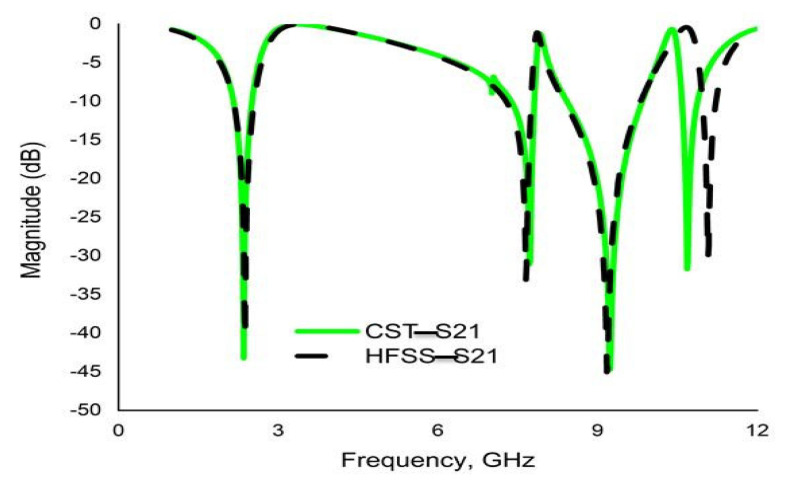
Comparison of transmission coefficients $\left(S_{21}\right)$ simulated by CST and HFSS.

**Table 1 materials-16-01137-t001:** Geometric values of the unit cell configuration.

Parameter Symbol	Size (mm)	Parameter Symbol	Size (mm)
L	11	g	0.40
B	11	R_1_	4.30
c	10.40	R_2_	3.80
d	10.40	r_1_	3.3
G	0.50	r_2_	2.8
e	0.75	r_3_	1.5
t	0.50	r_4_	0.75

**Table 2 materials-16-01137-t002:** Response of transmission coefficient (S_21_) for designs steps.

Design	Resonance Frequency (GHz)	Structural Composition	Magnitude (dB)	Band
Design-1	2.44, 8.67, 10.84	SRR with single octagon	−40.41, −44.48, −31.43	S-, X-
Design-2	2.51, 8.55, 9.50, 11.02	SRR with double octagons	−38.91, −39.09, −37.82, −19.29	S-, X-
Design-3	2.50, 8.57, 9.49, 11.03	SRR with three octagons	−39.83, −39.84, −37.88, −19.16	S-, X-
Design-4	2.51, 8.58, 11.05	SRR with three attached octagons	−42.69, −37.28, −15.73	S-, X-
Final Design	2.35, 7.72, 9.23, 10.68	SRR with three attached OSRRs	−43.23, −31.05, −44.58, −31.71	S-, C-, X-

**Table 3 materials-16-01137-t003:** Comparison of previously published MMs with the MM proposed in the present study.

Author Name & Ref.	Shape of MM	Size (mm^2^)	Substrate Material	Metamaterial Type	EMR (*λ*/L)
Islam et al. [46]	H	30 × 30	FR-4	LHM	3.65
Zhou et al. [47]	Double Z	8.5 × 8.5	FR-4	LHM	4.83
Islam M et al. [48]	SRR	16 × 12	FR-4	NRI	5.35
Hossain et al. [49]	Double dumbbell	9 × 9	FR-4	ENG	11.5
Proposed MM	Power tiller wheel	11 × 11	RT6002	ENG	11.61

**Table 4 materials-16-01137-t004:** Deviation of frequency for various arrays of unit cells.

Resonance Peak (GHz)	Unit Cell	1 × 2 Array	2 × 2 Array	Max. Shift of Frequency
First	2.35	2.57	2.57	0.22
Second	7.72	7.60	7.60	0.12
Third	9.23	9.47	9.48	0.25
Fourth	10.68	10.36	10.36	0.32

## Data Availability

All data are available within the manuscript.

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
