# Peer review of "An Innovative Compact Split-Ring-Resonator-Based Power Tiller Wheel-Shaped Metamaterial for Quad-Band Wireless Communication"

_materials, 2023, doi:10.3390/ma16031137_

Round 1
Reviewer 1 Report
In this work, a quad-band power tiller wheel shaped ENG metamaterial was designed and its properties were simulated comprehensively. The main achievement of this work is the four good resonance peaks covering S-, C-, and X-bands. According to me, this article is to some extent too long, and some simulation parts such as surface current, Magnetic/electric field distributions are not necessary. Some other issues are addressed below:
1. The subscript and superscript in the text should be modified.
2. L.90-95 and Fig. 1a: There are many mistakes in this part. For example: where is R2 in Fig. 1a; the radius of the second octagon is r1= 3.3 mm and r1= 2.8 mm??; the arrow marks in Fig.1a are not accurate.
3. Fig. 3, according to the following discussion in Fig. 4, the substrate used in Fig. 3 should be RT 6002, this should be mentioned in the discussion of Fig. 3.
4. Have the authors tried to fabricate the designed device? This will add many points to this work.
5. Relevant works on antennas working on S- and C-bands are recommended. Journal of the European Ceramic Society 41 (10), 5170-5175, and Journal of the European Ceramic Society 41 (9), 4835-4840
Author Response
As attached.

Reviewer 2 Report
1. Line 89,The copper (annealed) with thermal conductivity 5.96*107 Sm-1,here the thermal conductivity should be Electrical Conductivity.
2. The format of Equations in the paper should be readjusted.
3. The input and output ports and parameter value of each component should be marked in Figure 11.
4. The spelling and grammar of the full text should be checked.
5. Please cite the following articles (a) in the introduction and (b) in the ADS electric circuit simulation part:
(a) Yunsheng Guo, and Ji Zhou. Total broadband transmission of microwaves through a subwavelength aperture by localized E-field coupling of split-ring resonators. Optics Express, 2014, 22(22): 27136-27143.
(b) Yunsheng Guo, and Ji Zhou. Dual-band-enhanced transmission through a subwavelength aperture by coupled metamaterial resonators, Scientific Reports, 2015, 5(8144):1-5.
Author Response
As attached.

Reviewer 3 Report
The revised paper can be accepted for publication.
Author Response
Thank you for the recommendation.
Reviewer 4 Report
Review manuscript ID: materials-1951141
Type of manuscript: Article
Title: An Innovative Compact Split Ring Resonator Founded Power Tiller Wheel-Shaped Metamaterial for Quad Band Wireless Communication
Authors: Md. Salah Uddin Afsar, Mohammad Rashed Iqbal Faruque*, Sabirin Abdullah, Mohammad Tariqul Islam
Submitted to the section: Advanced Composites, https://www.mdpi.com/journal/materials/sections/adv_composites, Metamaterials and Their Applications https://www.mdpi.com/journal/materials/special_issues/metamater_application
Date: 8 November 2022
The article describes an experimental setup of a particular material setting and it has applications in the field of a wireless communication system. Although the introduction is sufficiently extensive by citing many previous studies in the literature, it seems that the novel contribution is not expressed explicitly. Some readers could guess that this is given in the final paragraph of the introduction but the sentences are vague and hard to understand, at least for general audiences, but I have a strong feeling that many specialists will have an easy time understanding too. Anyway, the following remarks and minor corrections might be useful for improving the manuscript.
* Sometimes the authors wrote the Greek letters in Greek letters but other times in Latin characters. Please write consistently.
* Line 21: insert a comma.
* Line 25: avoid a new sentence with "And".
* Line 27: environments
* Line 41: Nowadays
* Line 42: microwave
* Line 71: space needed
* Line 76: two
* Line 81--90: reword and improve.
* Professor Khiew Poi Sim from Nottingham Semenyih (UKM alumnus) and his collaborators, as well as Professor Janet Lim Hong Ngee from UPM Serdang and her collaborators, are working and have worked on material-related research. Please cite their works accordingly, if you find them relevant to your paper of course.
Author Response
As attached.

Round 2
Reviewer 1 Report
This manuscript can be accepted for publication.
Author Response
Thank you for the recommendation.
Reviewer 4 Report
Second review Manuscript ID: materials-1951141 Type of manuscript: Article
Title: An Innovative Compact Split Ring Resonator Founded Power Tiller Wheel-Shaped Metamaterial for Quad Band Wireless Communication
Authors: Md. Salah Uddin Afsar, Mohammad Rashed Iqbal Faruque *, Sabirin Abdullah, Mohammad Tariqul Islam Journal: Materials Submitted to section: Advanced Composites, Metamaterials and Their Applications Date: 27 November 2022 Thank you for revising the manuscript. Here are some remarks. * The explanation in the caption of each figure is still minimal and hardly communicates to the readers what you would like to convey. * The type of the manuscript is an article. We wondered why the authors called it a scientific report. I understand the latter falls in a different category among the types of documents published by MDPI. * depicted in * For a beautiful mathematical display, consider typesetting using LaTeX. Some formulas are rather ugly, such as (4). * coordinate * The figure uses x, y, and z, but the text uses X, Y, and Z. Be consistent! * Parameter symbols should be written in math mode, and not in text mode. Thus, LaTeX can do a good job for that. * What is CST? * Use and instead of ampersand. * The remark regarding space is ignored, e.g., Lines 147, 155. * What is NRW? * Design or design? Be consistent! * The lone bracket in (1) should be removed. * demonstrated in * Why are there large spaces in Line 185? * What is the difference between design-4 and design-4 final? This is confusing. * Some mathematical equations contain vector fields, but the authors did not make a distinction about this matter. This is a very serious mathematical mistake. * The permittivity in (14) is not written in Greek epsilon. That is "in" in the set theory notation. * correction factor * What is MM? ​* The caption in Figure 12 was cut down. Please correct it. * ​Each caption is missingu period.
Author Response
As attached.
